# The Impact of COVID-19 on the Lifestyles of University Students: A Spanish Online Survey

**DOI:** 10.3390/healthcare10020309

**Published:** 2022-02-05

**Authors:** Cayetana Ruiz-Zaldibar, Laura García-Garcés, Ángel Vicario-Merino, Noemí Mayoral-Gonzalo, Marta Lluesma-Vidal, Montserrat Ruiz-López, David Pérez-Manchón

**Affiliations:** 1Department of Nursing, Faculty of Health and Education, University of Camilo José Cela, 28692 Madrid, Spain; crzaldibar@ucjc.edu (C.R.-Z.); avicario@ucjc.edu (Á.V.-M.); nmayoral@ucjc.edu (N.M.-G.); mrlopez@ucjc.edu (M.R.-L.); dpmanchon@ucjc.edu (D.P.-M.); 2Department of Nursing, Faculty of Health Sciences, Universidad Cardenal Herrera-CEU, CEU Universities, 46115 Valencia, Spain; marta.lluesma1@uchceu.es

**Keywords:** COVID-19, lockdown, lifestyles, students, university, well-being, gender

## Abstract

The present study aimed to investigate the perceived changes in lifestyle behaviors among Spanish university students during COVID-19-related confinement. An observational, descriptive, and cross-sectional survey study was conducted during April 2020. Sociodemographic and anthropometric data were then obtained. The FANTASTIC questionnaire was used to assess the lifestyles of the 488 participants who took part in the study. Of the participants, 76.3% were female. Overall, the lifestyles of university students significantly deteriorated during the period of confinement caused by the COVID-19 pandemic in Spain. University female students were especially affected compared to their male fellows (*p* = 0.010). For women, social and family relationships (*p* < 0.001), personality (*p* < 0.001), interior (*p* < 0.001), and career (*p* < 0.001) were the aspects that worsened during confinement. For men, lack of physical exercise (*p* < 0.001), social and family relationships (*p* < 0.001), and career (*p* = 0.002) were affected to a greater extent. In both cases, confinement was a protective factor against the consumption of tobacco, toxins (*p* < 0.001), and alcohol (*p* < 0.001). Gender (*p* = 0.008) and obesity (*p* = 0.044) were the two factors that most affected the change in the score of the FANTASTIC questionnaire. Spanish university students’ lifestyles worsened during the lockdown caused by the COVID-19 pandemic, especially those of women, who were the most affected. Some aspects, such as those related to social and emotional behaviors, were deeply affected, while confinement could be a protective factor against previous toxic habits.

## 1. Introduction

The world is experiencing one of the most important global challenges in infectious disease management in recent times [1]. This is the case of the coronavirus pandemic (COVID-19) caused by the severe acute respiratory syndrome coronavirus 2 (SARS-CoV-2), which is causing millions of deaths worldwide [2]. The negative evolution towards complex lung diseases, generalized organ swelling and death, as well as the high expansion and contagion capacity attributed to this virus, led the World Health Organization (WHO) to declare the state of a world pandemic in March 2020 [3,4,5]. This caused a wave of confinements around the world as a preventive measure.

Spain declared a state of alarm on 14 March 2020, with 5753 cases detected and 136 deaths [6,7]. The Spanish lockdown was one of the most restrictive in the entire European Union. The Spanish population spent 49 days at home with restrictions on mobility, as all exits from their home were forbidden, except for basic food items and medicine shopping, work-identified essential areas, or transfers to hospitals [8,9].

Social distancing and isolation caused by the obligation to stay at home had an impact on people’s lives by modifying their habits [10], as well as causing abrupt and radical changes in lifestyles [11]. Lifestyles determine health by integrating a set of habits and behaviors that modulate our daily lives [12]. A lifestyle is a way of living that could be either healthy or unhealthy depending on personal behavioral choices. The promotion of healthy lifestyles serves to maintain or enhance the level of well-being, self-actualization, and fulfillment of an individual [13].

University students are a vulnerable population group in terms of the adoption of risky health behaviors. What is particularly remarkable about this period is the insufficient physical activity and sedentary behavior, poor diets with increases in sugar, fat, and sodium intake, and the tendency to skip breakfast. Although students have the knowledge of a balanced diet, these modifications could be associated with them living independently from their parents, the translation into cooking and eating practices, changes in body mass, and a lack of time or monetary concerns [14].

College is also a critical time for the adoption of unhealthy habits, such as smoking, drug consumption, drinking alcohol, or poor sleeping habits [15,16]. These patterns are acquired via the replication of students’ social environments, especially peers’ attitudes and behaviors, which could promote the adoption of these unhealthy lifestyles [14,17].

Mental health is at risk during the university period as students experience stress to achieve success in their studies, to be satisfied with their career, anxiety, depression, and academic concerns [18]. It is also a time of creating new relationships and emotional adjustments, changes in mind, and social relationships [17]. The university stage is also a critical period for personality since interpersonal confidence, social and emotional skills, as well as self-esteem are formed and can favor personal well-being [19].

A university is a context that influences the lifestyles of students [14] and is an active agent in health promotion [12]. However, the lockdown entailed the closure of universities and the lack of face-to-face classes led to a change in the well-being of students [20,21] at an academic level, in their social lives [22,23] and in their future prospects [24]. During confinement, Spanish universities experienced an unprecedented shift from traditional classroom education to online education. This involved a change in teaching methodologies, the adaptation of skills with new communication channels, the adaptation of assessment methods, the use of information and communications technology, workloads, and performance levels [24,25,26]. In addition, many students returned home, which meant a change in their places of residence [27]. Finally, isolation had a great impact on people’s physical and mental health, especially described during the current pandemic [28].

All these circumstances, added to the critical stage that this population was at, had an impact on the well-being of students. However, in addition, these consequences in well-being were not homogeneous in the university population. Some studies have shown that females presented a higher perception of danger to COVID-19 than males, and differences in mental health, nutrition, and physical activity were observed [29].

Since the declaration of the pandemic, numerous studies had been carried out on the effects of the pandemic on the well-being of the general population, but not so many have performed the same for university students. Some studies have investigated the effects on certain aspects of university students’ lifestyles, specifically mental health [30,31], physical exercise [32,33], diet [34], sleep [35,36] or a combination of various aspects [37]. However, there are few studies that have evaluated key factors related to the determinants of health, such as social and family relationships, drug use, or job satisfaction. We have not found any study evaluating a change in overall functioning perceived by Spanish university students during the COVID-19 pandemic related to relationships with family and friends, physical exercise, nutrition, tobacco, drug and alcohol consumption, sleep/stress, interior and work/type personality. The main objective of this study was to determine the perceived impact on the well-being of Spanish university students during the confinement caused by COVID-19 and analyze this change between women and men.

## 2. Material and Methods

### 2.1. Study Design and Setting

This is an observational, descriptive, and survey study carried out at the Department of Nursing of the (blinded for review) University. To obtain data on the lifestyles of the Spanish university population during the COVID-19 pandemic, a web survey was used through Microsoft Forms.

The survey was conducted from the 11th to the 25th April 2020 by using an online platform, accessible from any device with an internet connection, such as smartphones, computers, or tablets. The survey was shared through institutional emails, institutional and private social networks (Twitter, Facebook, and Instagram) and WhatsApp. Based on other studies [10], this method is effective for the research objectives since it facilitates the dissemination of the survey questionnaire during a period in that, due to the pandemic, there are many territorial restrictions.

This study complies with the STROBE (strengthening the reporting of observational studies in epidemiology) cross-sectional reporting guidelines (Appendix A).

### 2.2. Sample Size Calculation and Participants

The calculation of the sample size and power was performed using a sample size calculator from Qualtrics, as previously reported in other studies [38,39]. Considering a 95% confidence interval, a margin of error of 5% and a population size of 1,309,791 registered Spanish university students [40], the minimum number of cases required for a statistical power of 95% was 385 cases. We adjusted this sample size to accommodate potential losses. The academic literature [41] recommends adding 10% to the initial calculation, so we set the estimated loss ratio to 10%. Based on the data, the adjusted sample size was 424 participants.

Participants were included in the study based on the following eligibility criteria:

Inclusion criteria:Be 18 years of age or older.Be enrolled in a Spanish university.Online acceptance of informed consent.

Exclusion criteria:Having a serious disability or pathology that limited or conditioned life habits.Be a graduate student.

### 2.3. Data Collection

Due to the special circumstances of confinement and the urgency of obtaining the information, the sample carried out was of convenience, being applied to groups of students of which the researchers were professors, as well as students of colleagues from other Spanish universities. The sample was obtained from a group of 40 Spanish universities and a total of 488 students.

The study was carried out during the state of alarm decreed in Spain due to the COVID-19 pandemic, which made it impossible to carry out the questionnaire in person. The questionnaire was distributed virtually through a platform designed for conducting surveys (Microsoft Forms). This format favored the sincerity of the participants by avoiding the possible bias of feeling judged by the person administering the questionnaire. With the aim of knowing the change perceived by the participants in their lifestyles, they were asked through the survey platform to answer the following questionnaire twice: first in reference to their lifestyle before the COVID-19 pandemic (January 2020) and then at the time of confinement (April 2020).

### 2.4. Outcome Measures

Sociodemographic information was obtained related to age and sex, university studies (the university where you are enrolled, university degree and city where you reside), confinement (city of confinement, people with whom you live before and during confinement, characteristics of the dwelling, and number of departures from the home per week during the confinement) and COVID-19 (positive infection or suspected infection in the participant or partner/s). Additionally, self-reported height and body weight were included to calculate the body mass index (BMI) (Kg/m^2^) of the participants.

The lifestyle variable was measured with the validated FANTASTIC questionnaire [42]. This instrument was designed by Wilson and Ciliska [43] to measure the lifestyle of a population and has been validated in different countries, such as Mexico, Colombia, Brazil, and Spain, with a version available in Spanish and English. The current investigation used the Spanish version. FANTASTIC has been validated in different contexts, such as in the working population, students, or general consultation patients [42,44,45].

The FANTASTIC questionnaire consists of 25 closed items that explore lifestyles, divided into the following 9 dimensions: family and friends, physical activity, nutrition, tobacco and other toxic consumption, alcohol consumption, hours of sleep, use of seat belts, presence of stress, personality type, positive thoughts, anxiety, depression, and satisfaction with the tasks performed. The evaluation of the answers was carried out using a Likert-type scale, which measures the degree of opinion or behavior regarding each question asked, attributing a score that ranges from 0 to 4, where 0 corresponds to never or almost never and 4 to always or almost always, depending on the nature of the question. The maximum total possible score is 100 points, and the interpretation of the data in relation to healthy lifestyles in general is as follows: from 85 to 100 points, a lifestyle is considered as fantastic, from 70 to 84 to be considered good, from 60 to 69 to be considered average, from 40 to 59 to be considered low, and up to 39 points to be considered dangerous. The reliability of the questionnaire in its validated version in Spanish, measured through Spearman’s test–retest correlation, gave very good reliability. (r = 0.81, *p* = 0.01) [42].

### 2.5. Data Analysis

The analysis was performed with IBM SPSS Statistics v.21 software (IBM, Madrid, Spain).

Statistical analysis of sociodemographic data was performed using the mean and standard deviation (SD) or medians, as well as minimums and maximums for the quantitative variables according to their distribution. For qualitative or categorical variables, measures of frequency and percentages were used. The parametric properties of the FANTASTIC questionnaire scores were analyzed to determine the normality of their distributions. A Student’s paired *t*-test was used to analyze the differences between pre- and during quarantine. A Student’s unpaired *t*-test analyzed the differences between males and females. A chi-square (χ^2^) test was used to determine the significance of differences in the frequency distribution of FANTASTIC categories and Fisher’s exact test was used when any of the expected values were less than 5. To study the association between sociodemographic variables and lifestyles, a logistic regression model with sequential backward adjustment was fitted. The dependent variables included were the mean differences between pre-confinement and during confinement (FANTASTIC questionnaire score), and the independent variables were the sociodemographic data. A final model was constructed with the variables that were significantly associated. The effect size was found based on Cohen’s d [46]: 0–0.3 low, >0.3–0.8 moderate, and >0.8 big effect size. 

Those questionnaires that were not completed were excluded from the analysis. The results were considered statistically significant when *p* < 0.05.

### 2.6. Ethical Considerations

The project was approved by the Research Ethics Committee of the Camilo José Cela University (code: 06_CEI_2020). The study was conducted in full agreement with national law (Law 14/2007 on Biomedical Research and Law 15/1999 on Protection of Personal Data) and the Declaration of Helsinki (2000). All participants were informed about the study and required to accept the informed consent before participating in it.

Their participation was anonymous and voluntary, and students could withdraw from the study without any consequences at any time. The participants completed the questionnaire directly on the institutional Microsoft Forms platform, where the information was kept in a private storage of the university to which only the researchers had access. Participants’ personal information was anonymized to maintain and protect confidentiality. We did not ask for participants’ names or email addresses. The anonymous nature of the web survey does not allow sensitive personal data to be traced in any way.

## 3. Results

A total of 495 questionnaires were received. Of these, seven questionnaires were discarded because they were incomplete. Consequently, 488 questionnaires were analyzed. The vast majority of the sample gender was female, with 73.6% (n = 359), and the total sample had a median age of 21 years. Most men and women had a normal weight according to their BMI (73.4%), with significant differences in the proportion of women with low weight (χ^2^ = 8.71, *p* = 0.003) (see Table 1). Most of the Spanish regions were represented. In terms of areas of study, women studied more health sciences compared to men (χ^2^ = 25.85, *p* = 0.001), while men studied more engineering and architecture careers than women (χ^2^ = 39.02, *p* = 0.001). As for the participants with work activity, there was a higher proportion of men who worked both before (χ^2^ = 7.5, *p* = 0.006) and during (χ^2^ = 4.8, *p* = 0.028) the quarantine, with women losing the most jobs during the confinement. In general, statistically significant differences were found at work and in weekly hours, since during the quarantine there were fewer people working (χ^2^ = 96.3, *p* < 0.001) and those who worked performed more hours (*p* < 0.001).

Similarly, there were differences regarding the change of address (see Table 2); most of the participants went from living with their roommates (41.6% vs. 4.1%, χ^2^ = 20.1, *p* < 0.001) to living with their parents (48.8% vs. 86.5%, χ^2^ = 0.3, *p* < 0.001). Regarding the data on COVID-19, 92.4% had no confirmed infection or suspected disease or lived with people with symptoms. Of the participants, 59% affirmed not leaving home during mandatory confinement.

The global score of lifestyles measured with the FANTASTIC questionnaire worsened during confinement in women (*p* < 0.001) with a moderate effect size (*d* 0.32) (see Table 3). If we focus on the items in each dimension, the questionnaire also reflects that the effect of confinement was different for men and women (see Table 3). Active exercise of at least 30 min decreased significantly and with a large effect size in men (*p* < 0.001, *d* 0.99), while in women, it increased significantly (*p* = 0.008), with a small effect size (*d* 0.15). Women consumed significantly more drugs (with and without prescription) during confinement, with a large effect size (*p* < 0.001, *d* 1.58), while men decreased their drug use (*p* < 0.001, *d* 0.33) with a moderate effect size. For women, during confinement, following a balanced diet (*p* = 0.045, *d* 0.22) and maintaining an ideal weight (*p* = 0.013, *d* 0.08) worsened, but excess consumption of sugar, salt, fat, and junk food was reduced (*p* = 0.050, *d* 0.10), although with small effect sizes. In men, no statistically significant differences were observed in these items. On the other hand, those related to adequate nighttime sleep (7 to 9 h) with small effect sizes and significant stress episodes improved significantly in both (men *p* = 0.041, *d* 0.32 and women *p* = 0.035, *d* 0.08), with a moderate effect size specifically in men.

By category (fantastic, good, moderate, low, and worrying), significant differences in the increase in participants who reached the fantastic level of lifestyle after confinement were found (men χ^2^ = 42.3, *p* < 0.001 and women χ^2^ = 126.1, *p* < 0.001), fewer participants reached the good category (men χ^2^ = 22.4, *p* < 0.001 and women χ^2^ = 38.8, *p* < 0.001), there was a higher proportion in the moderate category (men χ^2^ = 8.7, *p* = 0.006 and women χ^2^ = 21.6, *p* < 0.001) and a higher proportion in the low category for women (χ^2^ = 58.3, *p* < 0.001). During the confinement there were two women who fell into the group of worrying in their lifestyles (for more details see Table 4).

Regarding the nine dimensions of the questionnaire, family and friends relationships (men *p* < 0.001, *d* 0.39, and women *p* < 0.001, *d* 0.75) as well as career (men *p* = 0.002, *d* 0.33, and women *p* < 0.001, *d* 0.39) in both genders worsened significantly with moderate effect sizes. By gender, physical exercise for men decreased significantly with a moderate effect size (*p* < 0.001, *d* 0.52), while in women, the personality dimension worsened significantly with moderate effect sizes (*p* < 0.001, *d* 0.32), as did the interior dimension (*p* < 0.001, *d* 0.36). On the contrary, toxic habits improved by significantly reducing the consumption of tobacco and toxins (men *p* < 0.001, *d* 0.21 and women *p* < 0.001, *d* 0.15) with small effect sizes and alcohol consumption (men *p* < 0.001, *d* 0.51 and women *p* < 0.001, *d* 0.24) with a moderate effect size in men. No differences were found in diet.

In general, confinement affected more women than men (*p* = 0.010, *d* 0.29) with a small effect size (see Table 5). For women, the aspects that were most affected were relationships with family and friends (*p* = 0.002, *d* 0.29) with a small effect size and personality (*p* < 0.001, *d* 0.32) as well as interior (*p* < 0.001, *d* 0.36) with a moderate effect size. For men, it affected the performance of physical exercise more negatively (*p* < 0.001, *d* 0.34) with a moderate effect size. The decrease in alcohol consumption affected more men (*p* < 0.001, *d* 0.41) with a moderate effect size (see Table 5).

The logistic regression model with sequential backward adjustment identified that the two factors that most affected the change in the score of the FANTASTC questionnaire were gender (*p* = 0.008) and BMI, where obesity was the most influential in its categories (*p* = 0.044) (see Table 6).

## 4. Discussion

To date, this is the first study evaluating the perceived effect of confinement on the lifestyles of university students in Spain during lockdown. Our findings indicate that, in general, the lifestyles of university students worsened significantly during confinement caused by the COVID-19 pandemic in Spain. This is particularly the case for female university students, who were the most affected in comparison to their male counterparts. For women, aspects related to social and family relationships, positive thinking, or feelings of anger worsened during confinement. For men, a lack of physical exercise was affected to a greater extent. In both, confinement was a protective factor against the consumption of tobacco, toxins, and alcohol. However, for women, both prescription and non-prescription drug abuse worsened.

The sample of this study was made up mostly of women (73.4%). Half of the sample were studying health sciences, while the men who made up the sample (26.6%) studied engineering or architecture. These statistics are in line with current trends in which women are probably more health-conscious, more interested in participating in these studies, and study more health-based science careers, while men participate less and study for university degrees in engineering [47]. The anthropometric characteristic studied from the BMI is in line with national and global data [48,49], where the majority category is that of normal weight (BMI 18.5–24.9 kg/m^2^); however, there was a high prevalence of overweight participants compared to the total (16.4%). Although obesity accounts for 2.9%, it is one of the factors that most affected the change in scores shown in the FANTASTIC questionnaire. The underweight factor was represented mainly by women (9.5% women vs. 1.6% men). According to the 2017 National Health Survey, the prevalence of underweight in women aged between 18 and 24 years old was 12.7%, compared to 3.5% in men, which indicates that our sample is close to the reality of the Spanish university population [50]. In this sense, being underweight continues to be perpetuated as a synonym for beauty, especially among university women, which can lead to future health problems [51].

The socioeconomic contexts of individuals and countries impact health. In this pandemic, both are at risk since, as the results indicate, most of the students who worked lost their jobs. This was especially the case for women, who in fact lost more jobs. These results are consistent with other studies where Spanish women were slightly more likely to lose their jobs than men, and those who remained employed were more likely to work from home [52,53]. This situation could be one of the reasons that justify higher feelings of anger and lower positive thoughts in women. Losing a job and having no expectations of finding another in a short period of time could lead to these changes in the personality and interior dimensions.

For females, there was a higher percentage that descended from a “Good” standard of living to levels classified as “Moderate,” “Low,” and “Worrying,” according to the classification of global scores found in the FANTASTIC questionnaire. This implies the presence of possible health risk indicators [44]. In contrast, there was a higher proportion of men who upgraded to “Fantastic”. This highlights the heterogeneity in the effects of confinement for students, especially between men and women, which may be based on a detrimental or beneficial factor.

The results mainly indicated a worsening of psychological and relationship factors during confinement for women. These results are in line with those presented by a study carried out in Spain with a sample of 2070 individuals aged between 18 and 75 years old, in which the psychological response of the Spanish population to the COVID-19 crisis was evaluated. The results showed that women had greater symptoms of depression and anxiety than men. In addition, it concluded that the age group with the most symptoms of depression (42.9%) and anxiety (34.6%) was the youngest (18–24 years old) [54]. What is especially striking in our study are the results of less honest, open, and clear communication with family and friends, provided and received affection, obtained the emotional support needed, and less positive thinking. Social isolation, the inability to continue with usual routines and the impediment to carrying out life projects (trips, ceremonies, parties, and meetings, among others) typical of confinement, can promote feelings of anger or aggressiveness. These results are in agreement with studies carried out in other contexts, such as the USA, Indonesia, Thailand, or Taiwan, where it was found that university students increased their levels of stress, depression, and suicidal thoughts [55,56]. Some authors have indicated that these psychological consequences have been seen in previous outbreaks, such as severe acute respiratory syndrome in cities in both China and Canada in 2003, in addition to Ebola in some African countries in the year 2014. Several studies indicate that confinement, loss of habitual routine, and reduced social and physical contact with others are frequently associated with feelings of boredom, frustration, and a feeling of isolation from the rest of the world [57]. These findings are in line with other studies where stress and anxiety levels increase with age and responsibility; however, they affirm that these levels decrease throughout the days of confinement [58].

Men decreased their practice of exercise during confinement, while women slightly increased it. These results contrast with other studies, such as those of Sánchez-Sánchez et al. [59], where it was found that the lack of physical activity during confinement was more notable in women than in men. The difference in exercise patterns between women and men probably affected these results. It is likely that women were able to adapt their physical exercise with synchronous online activities directed by professionals, such as yoga, Pilates, or Zumba, among others, while men, who are more used to collective exercises, had certain difficulties in continuing with their usual physical exercise.

In general, those aspects related to social behaviors, such as alcohol or tobacco consumption, underwent improvements. This leads us to think that confinement may be a protective factor when it comes to students’ toxic habits. The first reason a student begins or continues to use drugs and alcohol is the availability of and access to illicit drugs. Most students report that they have easy access and opportunity to consume cannabinoids and prescription stimulants [60,61]. The second reason is the decrease in parents’ capacity to exert a direct protective effect through the supervision of the whereabouts and activities of their children, especially for students living outside their home [62]. However, these two factors have radically changed during confinement and may influence the sense of belonging to a group. Students who smoke identify themselves as social smokers [63], and there is a relationship between drug, alcohol, and tobacco use [64,65,66]. Nevertheless, the fact is that the consumption of prescription and non-prescription drugs increased in women. This fact is probably related to other factors identified in the study, such as the fact that confinement affected women more in their psychosocial area.

Insomnia and sleep disorders are common problems among university students [67,68]. Night preferences [69,70], social networks and the use of the Internet or mobile phones [71] have been defined as harmful factors for sleep, while the practice of physical activities is beneficial [72]. In our results, students increased their hours of sleep and reduced stressful events. Isolation, the provision of more time, the maintenance of exercise practice or living with parents again probably contributed to this improvement. These results are in accordance with the study of Romero-Blanco et al. [32], where it was found that nursing students slept more hours during lockdown. However, according to this study, the quality of sleep worsened.

### Strengths and Limitations

Faced with the threat of future outbreaks of the COVID-19 pandemic or new pandemics worldwide, this study shows that the healthy lifestyles of the university population may be harmed. University women, in particular, are the most affected during confinement. In this sense, it is essential to develop strategies that favor the social and psychological factors that negatively affect the health of students with a gendered perspective. This implies necessary strategies for health promotion with a gendered approach that addresses the differences between women and men in an equitable manner [73,74]. The sample size of this cross-sectional survey was relatively large, having exceeded the initial number of the sample size calculation. The included participants had generalizable characteristics of the Spanish university population. Participants were recruited from most of the Spanish regions and were studying at various Spanish universities. An important aspect of the study is the contrasting measurement of outcome factors before and during the pandemic.

However, general statements and interpretations of the current findings should be made with caution, as some of the differences that were found are clinically small. One of the main limitations of the study was the self-reported data, which implies the chance of reporting bias. Another factor is that there were no reporting data on the participants’ socioeconomic status, which could be important for the analysis.

## 5. Conclusions

The findings showed that Spanish university students have seen their healthy lifestyles diminished during confinement by COVID-19. Females reported more worsening in their lifestyles than males did. For women, psychological and social factors were the most affected aspects, while for males, it was exercise. Adapted strategies should be developed to try to mitigate its impact with a gendered perspective. Aspects related to the consumption of toxics and drugs were diminished thanks to confinement, which could be a protective factor. At this point, for females, drug consumption worsened. In this sense, this research has revealed the need to develop interventions that promote the adoption of healthy lifestyles by the Spanish university population during confinement for COVID-19.

## Figures and Tables

**Table 1 healthcare-10-00309-t001:** Sociodemographic characteristics of participants.

Sociodemographic Characteristics	Total Sample (n = 488)	Male (%) 129 (26.4)	Female (%) 359 (73.6)	*p*-Value
Age (Years)				
Median (min–max)	21 (18–54)	21 (18–54)	21 (18–49)	0.205
BMI (kg/m^2^)				
<18.5 underweight (%)	36 (7.4)	2 (1.6)	34 (9.5)	χ^2^ = 8.71, *p* = 0.003
18.5–24.9 normal (%)	358 (73.4)	97(75.2)	261 (72.7)	χ^2^ = 3.02, *p* = 0.583
25–29.9 overweight (%)	80 (16.4)	28 (21.7)	52 (14.5)	χ^2^ = 3.61, *p* = 0.057
>30 obesity (%)	14 (2.9)	2 (1.6)	12 (3.3)	χ^2^ = 1.09, *p* = 0.373 ^*^
Region of Spain Where You Study				
Madrid (%)	111 (22.7)	31 (24)	80 (22.3)	χ^2^ = 2.99, *p* = 0.224
Valencia (%)	102 (20.8)	33 (25.6)	69 (19.2)	χ^2^ = 2.32, *p* = 0.227
Basque Country (%)	85 (17.4)	32 (24.8)	53 (14.8)	χ^2^ = 9.63, *p* = 0.008
Andalusia (%)	71 (14.5)	10 (7.8)	61 (17)	χ^2^ = 9.15, *p* = 0.010
Others (%)	119 (24.6)	23 (17.8)	96 (26.7)	χ^2^ = 4.09, *p* = 0.043
Study Areas				
Art and humanities (%)	22 (4.5)	4 (3.1)	18 (5)	χ^2^ = 0.81, *p* = 0.369
Science (%)	33 (6.8)	11 (8.5)	22 (6.1)	χ^2^ = 0.87, *p* = 0.352
Health science (%)	245 (50.2)	40 (31)	205 (57.1)	χ^2^ = 25.85, *p* = 0.001
Engineering and architecture (%)	73 (15)	41 (31.8)	32 (8.9)	χ^2^ = 39.02, *p* = 0.001
Social and legal science (%)	115 (23.6)	33 (25.6)	82 (22.8)	χ^2^ = 0.38, *p* = 0.539
Employment Pre-				
Yes (%)	133 (27.3)	47 (36.4)	86 (24)	χ^2^ = 7.5, *p* = 0.006
Hours per Week	(n = 131)	(n = 47)	(n = 84)	*p* = 0.490
Median (min–max)	20 (1–54)	20 (2–54)	17 (1–48)	
Employment During				
Yes (%)	54 (11.1)	21 (16.3)	33 (9.2)	χ^2^ = 4.8, *p* = 0.028
Hours per Week	(n = 49)	(n = 19)	(n = 30)	*p* = 0.221
Median (min–max)	30 (0–84)	30 (2–84)	30 (0–40)	
Have You Moved Since the Confinement?				
Yes (%)	266 (54.5)	64 (49.6)	202 (56.3)	χ^2^ = 1.7, *p* = 0.193
People You Live with Pre-				
Alone (%)	23 (4.7)	6 (4.7)	17 (4.7)	χ^2^ = 0.01, *p* = 0.969
With my parents (%)	238 (48.8)	71 (55)	167 (46.5)	χ^2^ = 5.80, *p* = 0.055 *
With my roommates (%)	203 (41.6)	46 (35.7)	157 (43.7)	χ^2^ = 5.14, *p* = 0.077 *
With my partner (%)	24 (4.9)	6 (4.7)	18 (5)	v = 0.39, *p* = 0.823 *
People You Live with During				
Alone (%)	17 (3.5)	4 (4.1)	13 (3.6)	χ^2^ = 0.44, *p* = 0.803 *
With my parents (%)	422 (86.5)	112 (86.8)	310 (86.4)	χ^2^ = 0.04, *p* = 0.893
With my roommates (%)	20 (4.1)	5 (3.9)	15 (4.2)	χ^2^ = 0.02, *p* = 0.882
With my partner (%)	29 (5.9)	8 (6.3)	21 (5.8)	χ^2^ = 0.02, *p* = 0.885
Symptoms of COVID-19				
Yes (%)	37 (7.6)	14 (89.1)	23 (6.4)	χ^2^ = 2.7, *p* = 0.102
People You Live with Present Symptoms of COVID-19				
Yes (%)	22 (4.5)	6 (4.7)	16 (4.5)	χ^2^ = 0.1, *p* = 0.927
Do You Leave Your Home?				
Yes (%)	200 (41)	55 (42.6)	143 (39.8)	χ^2^ = 0.3, *p* = 0.578
Times a Week	(n = 200)	(n = 55)	(n = 155)	
Median (min–max)	2 (1–17)	2 (1–14)	2 (1–17)	*p* = 0.952
Characteristic of Your Home				
Terraced house (%)	49 (10)	12 (9.3)	37 (10.3)	χ^2^ = 0.47, *p* = 0.790 *
Independent house (%)	91 (18.6)	22 (17.1)	69 (19.2)	χ^2^ = 0.29, *p* = 0.588
Flat without balcony (%)	110 (22.59)	29 (22.5)	81 (22.6)	χ^2^ = 0.36, *p* = 0.835 *
Flat with balcony (%)	238 (48.8)	66 (51.2)	172 (47.9)	χ^2^ = 0.40, *p* = 0.526

* Fisher’s exact test when any of the expected values <5.

**Table 2 healthcare-10-00309-t002:** Participants’ characteristics pre and during confinement.

Variables	Pre-Confinement (n = 488)	During Confinement (n = 488)	*p*-Value
People You Live With			
Alone (%)	23 (4.7)	17 (3.5)	χ^2^ = 141.2, *p* < 0.001
With my parents (%)	238 (48.8)	422 (86.5)	χ^2^ = 0.3, *p* < 0.001
With my roommates (%)	203 (41.6)	20 (4.1)	χ^2^ = 20.1, *p* < 0.001
With my partner (%)	24 (4.9)	29 (5.9)	χ^2^ = 300.1, *p* < 0.001
Employment			
Yes (%)	133 (27.3)	54 (11.1)	χ^2^ = 96.3, *p* < 0.001 *
Hours per Week Worked			
Median (min–max)	20 (1–54)	30 (0–84)	*p* < 0.001

* Fisher’s exact test when any of the expected values <5.

**Table 3 healthcare-10-00309-t003:** Differences by gender of global scores of the FANTASTIC questionnaire.

Variables	Male (n = 129)		*p*-Value	Effect Size	Female (n = 359)		*p*-Value	Effect Size
	Mean pre-conf. (SD)	Mean during conf. (SD)			Mean pre-conf. (SD)	Mean during conf. (SD)		
FANTASTIC Global Score	76.4 (±8.0)	75.5 (±10.4)	0.198	0.16	74.2 (±8.3)	71 (±11.5)	<0.001	0.32
FANTASTIC Dimensions and Items								
Family and Friends Honest, open, and clear communication I provide and receive affection I obtain the emotional support I need	9.9 (±1.9) 3.5 (±0.6) 3.3 (±0.8) 3.2 (±0.9)	9 (±2.7) 3.2 (±0.9) 2.9 (±1.1) 2.8 (±1.1)	<0.001 0.001 <0.001 0.001	0.39 0.39 0.42 0.40	10 (±1.9) 3.5 (±0.7) 3.3 (±0.8) 3.1 (±0.9)	8.2 (±2.8) 3.0 (±0.9) 2.6 (±1.2 2.6 (±1.2)	<0.001 <0.001 <0.001 <0.001	0.75 0.62 0.69 0.47
Physical Exercise Active exercise 30 min Relaxation and enjoyment of free time	6.4 (±1.5) 4.2 (±1.0) 3.2 (±0.9)	5.4 (±2.3) 3.5 (±1.5) 3.0 (±1.4)	<0.001 <0.001 0.106	0.52 0.99 0.17	5.2 (±1.8) 2.4 (±1.3) 2.8 (±1.1)	5.3 (±2.3) 2.6 (±1.4) 2.7 (±1.4)	0.452 0.008 0.064	0.05 0.15 0.08
Nutrition Balanced diet Daily breakfast Excess sugar, salt, fats, or junk foods You are an ideal weight	11 (±2.3) 3.1 (±0.9) 3.1 (±1.4) 1.6 (±0.9) 3.4 (±1.2)	11.2 (±2.4) 3.2 (±0.9) 3.2 (±1.4) 1.4 (±1.1) 3.4 (±1.2)	0.326 0.052 0.261 0.128 0.867	0.09 0.11 0.07 0.20 0.00	11.4 (±2) 3.1 (±0.8) 3.3 (±1.2) 1.7 (±0.9) 3.4 (±1.2)	11.3 (±2.2) 2.9 (±1.0) 3.3 (±1.3) 1.8 (±1.1) 3.3 (±1.2)	0.449 0.045 0.965 0.050 0.013	0.05 0.22 0.00 0.10 0.08
Tobacco and Toxics Tobacco consumption Drug abuse Coffee, tea, and cola beverages	9.7 (±2.5) 3.0 (±1.6) 3.5 (±1.0) 3.2 (±0.6)	10.2 (±2.2) 3.1 (±1.5) 3.8 (±0.8) 3.3 (±0.7)	<0.00 0.016 <0.001 0.009	0.21 0.06 0.33 0.15	9.9 (±2.1) 3.0 (±1.5) 4.7 (±0.8) 3.2 (±0.6)	10.2 (±1.8) 3.1 (±1.4) 3.7 (±0.4) 3.2 (±0.7)	<0.001 0.001 <0.001 0.438	0.15 0.07 1.58 0.00
Alcohol Weekly mean consumption Drink alcohol and drive	7.4 (±1) 3.6 (±0.8) 3.8 (±0.5)	7.8 (±0.5) 3.8 (±0.5) 4.0 (±0.1)	<0.001 0.001 <0.001	0.51 0.30 0.55	7.8 (±0.5) 3.9 (±0.4) 3.9 (±0.4)	7.9 (±0.3) 3.9 (±0.3) 4.0 (±0.1)	<0.001 0.049 <0.001	0.24 0.00 0.34
Relax, Security and Stress Sleeps 7 to 9 h per night Frequency of using the safety belt Important stress episodes	9.9 (±1.8) 2.8 (±1.2) 3.9 (±0.5) 3.2 (±1.0)	9.9 (±2.2) 3.0 (±1.3) 3.5 (±1.3) 3.5 (±0.9)	0.905 0.011 <0.00 0.041	0.00 0.16 0.41 0.32	9.2 (±1.9) 2.6 (±1.2) 3.9 (±0.3) 2.7 (±1.2)	9.2 (±2.4) 2.9 (±1.3) 3.5 (±1.3) 2.8 (±1.3)	0.731 <0.001 <0.001 0.035	0.00 0.24 0.42 0.08
Personality Feeling of urgency or impatience Competitiveness and aggressiveness Feelings of anger and hostility	7.9 (±2.1) 2.6 (±1.0) 2.4 (±1.0) 3.0 (±0.9)	8 (±2.6) 2.6 (±1.1) 2.7 (±1.1) 2.7 (±1.1)	0.470 1.00 0.001 0.018	0.04 0.00 0.29 0.29	8.0 (±2.1) 2.2 (±1.0) 2.9 (±0.9) 3.0 (±0.9)	7.2 (±2.8) 2.0 (±1.1) 2.8 (±1.1) 2.4 (±1.2)	<0.001 0.002 0.011 <0.001	0.32 0.19 0.10 0.57
Interior Thinks positively Anxiety and concern Depression	8.4 (±1.8) 3.1 (±0.8) 2.4 (±1.0) 3.4 (±0.9)	8.2 (±2.5) 2.8 (±0.9) 2.3 (±1.1) 3.2 (±1.0)	0.689 <0.001 0.171 <0.00	0.09 0.35 0.10 0.21	7.4 (±2.2) 2.8 (±1.0) 1.9 (±1.0) 3.1 (±1.0)	6.5 (±2.8) 2.2 (±1.1) 1.5 (±1.1) 2.7 (±1.2)	<0.001 <0.001 <0.001 <0.001	0.36 0.57 0.38 0.38
Career Satisfaction with the work and the activities Good relationships with those around you	6.2 (±1.4) 2.9 (±0.9) 3.6 (±0.6)	5.7 (±1.6) 2.5 (±1.0) 3.2 (±0.9)	0.002 <0.001 <0.001	0.33 0.42 0.52	6.0 (±1.3) 2.8 (±1.0) 3.5 (±0.6)	5.3 (±2.2) 2.1 (±1.8) 3.2 (±0.9)	<0.001 <0.001 <0.001	0.39 0.48 0.39

SD = standard deviation; effect size = Cohen’s *d* (0–0.3 low, >0.3–0.8 moderate and >0.8 big).

**Table 4 healthcare-10-00309-t004:** Differences by gender of categories of the FANTASTIC questionnaire.

Variables	Male (n = 129)		*p*-Value	Female (n = 359)		*p*-Value
FANTASTIC Categories Fantastic	Pre-conf. n (%) 18 (14)	During conf. n (%) 31 (23.3)	χ^2^ = 42.3 *p* < 0.001 *	Pre-conf. n (%) 34 (9.5)	During conf. n (%) 42 (11.7)	χ^2^ = 126.1 *p* < 0.001 *
Good	86 (66.7)	65 (50.4)	χ^2^ = 22.4 *p* < 0.001	226 (63)	163 (45.4)	χ^2^ = 38.8 *p* < 0.001
Moderate	22 (17.1)	24 (18.6)	χ^2^ = 8.7 *p* = 0.006 *	85 (23.7)	95 (26.5)	χ^2^ = 21.6 *p* < 0.001
Low	3 (2.3)	9 (7)	χ^2^ = 23.0 *p* = 0.804 *	14 (3.9)	57 (15.9)	χ^2^ = 58.3 *p* < 0.001 *
Worrying	0	0	NA	0	2 (0.6)	NA

NA = not applicable; * Fisher’s exact test when any of the expected values <5.

**Table 5 healthcare-10-00309-t005:** FANTASTIC global dimensions and score differences between males and females before and during confinement.

Parameters of FANTASTIC Questionnaire	Dif. Male (SD)	Dif. Female (SD)	*p*-Value	Effect Size
FANTASTIC Global Score	0.9 (±8.0)	3.2 (±8.7)	0.01	0.29
FANTASTIC Dimensions				
Family and friends	1.0 (±2.4)	1.7 (±2.5)	0.002	0.29
Physical exercise	0.9 (±2.2)	−0.1 (±2.5)	<0.001	0.34
Nutrition	−0.1 (±1.5)	0.1 (±1.7)	0.244	0
Tobacco and toxics	−0.5 (±1.2)	−0.3 (±1.1)	0.120	0.17
Alcohol	−0.4 (±0.9)	−0.1 (±0.5)	<0.001	0.41
Relaxation, security, and stress	−0.02 (±2.2)	0.0 (±2.3)	0.781	0
Personality	−0.1 (±2.4)	0.9 (±2.6)	<0.001	0.32
Interior	0.1 (±2.2)	0.9 (±2.3)	0.001	0.36
Career	0.4 (±1.6)	0.8 (±2.2)	0.081	0.21

Dif. = difference between pre- vs. during confinement; SD = standard deviation.

**Table 6 healthcare-10-00309-t006:** Factors associated with decrease in FANTASTIC score.

	Coef. (CI 95%)	*p*-Value
Gender	2.35 (4.09–0.62)	0.008
BMI (Kg/m^2^)		
18.5–24.9 normal weight (%)	2.34 (0.61–5.28)	0.119
25–29.9 overweight (%)	1.61 (1.78–5.00)	0.351
>30 obesity (%)	5.40 (0.14–10.67)	0.044
Pseudo R2 0.0235

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
