# Peer review of "The Impact of COVID-19 on the Lifestyles of University Students: A Spanish Online Survey"

_healthcare, 2022, doi:10.3390/healthcare10020309_

Round 1
Reviewer 1 Report
Manuscript ID: healthcare-1581851
Title: The Impact of COVID-19 on the Lifestyles of University Student: A Spanish Online Survey
Comments and Suggestions for Authors:
The topic and the idea of the study is valuable and corresponds very well to the contemporary challenges related to the changes caused by the covid-19 pandemic. Only selected elements of the manuscript should be improved to highlight this important topic and interesting results obtained by the Authors.
1. The title is clear and in line with the presented study, it explain the studied context presented in the manuscript.
2. Abstract is well constructed, clear and presents the most important elements of the study.
3. Key words are related to the topic, well selected, but could be developed by adding more terms which are crucial for presented aspects.
4. The issues in the section of Introduction are important and well selected in the context of the study, what is valuable. General background is clear. However there are some missed elements in my opinion.
In the first paragraph in the section of Introduction, the Authors present a general context of lock-down in Spain, pointing it as one of the largest restrictions in Europe with the negative impact on people. The information of the length of the confinement, which seems to be important to further emphasize the role and impact of this period on students reactions / feelings, must be added (lines: 50-51). It is crucial to understand / to confirm / the results of the study more deeply.
The university students are well characterized and the relations of their behaviors in the context of the changes in pandemic period are clear and present quite wide background. The references to current research in cited literature are appropriate.
The aim of the study is described synthetically and clear.
5. The only short term: Methods is not enough in my opinion as the title of the part 2 of the manuscript. This section, which is much developed and reach of information, should be called Material and Methods to better introduce the scope of presented information, or Methodology – this term also include both material and methods which are presented in this section.
The division into subsections is clear, and in most parts well prepared, also the scope of information on all of them is generally understandable.
The study design itself is well organized, also the proposed form of description of the following stages of the study is correct. However selected part of this section – presentation of participants – have some shortcomings/deficiencies, or the complementary data should be brought together. In my opinion, both subsection 2.2. Participants and 2.3. Sample Size Calculation are related to the presentation of participants, do not need to be presented separately. At the same time the sample size is sufficient and well argued.
The subsection 2.5. Outcome Measures focuses on very important information, which are presented very well in my opinion. The FANTASTIC method used by the Authors is acceptable and adequate to obtain sufficient results. The rules of measurement are clear. Also the Data Analysis are sufficient and described properly described in my opinion. The subsection 2.7. Ethical Considerations is also important part of the manuscript and well presented.
6. The results are presented in well organized order, including enough form of explanation. However, e.g. the tables are quite big, especially Table 3 – in its present form the part including FANTASTIC categories is somehow a bit mixed with the FANTASTIC Global Score and FANTASTIC dimensions & items - all parts are presented as a continuous form. If the Authors decide to keep all the data in one Table, it must include some horizontal lines which have to separate each of those 3 sections. Otherwise the data will be difficult to read and understand.
7. Discussion – this section is very well developed and includes many important observations, the whole description of presented aspects is discussed step by step and thus is logical and complete. This part of the study is very valuable, the Authors have noticed and discussed important relationships between the obtained results. The only weakness may be the amount/scope of other studies cited from other literature items – this reference should be a bit broader in my opinion.
8. The English language is generally understandable, but some sentences require a small proofreading to increase the quality of the presentation of this important topic.
Summing up, the topic, study design, description and discussion of results are valuable and represent high level. The above-mentioned parts with some weakness of presentation/description should be a bit improved. In my opinion, the manuscript can be published after this minor corrections.
Author Response
Thanks for all the comments and suggestions. We have made changes to the paper regarding your indications.

Reviewer 2 Report
The paper is relatively well-written and addresses an important problem. However, some clarifications on the methods, results and discussion and a careful proofreading are required. There are several incomplete sentences and grammar errors.
P.7, ln.246-7: it says "significant differences were found in the increase of participants who reach the fantastic level of lifestyles after confinement". It seems inconsistent with other findings. Table 3 shows decreases. Ln.250: it says "low in women". Shouldn't there be a category?
I could not find Figure 1.
Table 5 is not well explain and the methods to obtain it are not described.
P.10, ln. 312: it says "This fact could be related to the change to less healthy lifestyles of social context and the family environment". It seems to suggesst causality; however, the methods in the paper are only adequate to draw inferences of correlation. It seems to suggest lifestyle changes caused more obesity; however, the results only suggest obese people are more likely to experience lifestyle changes.
Table 3 shows more people goes from Good to Fantastic than to Moderate or Low in Males. The proportion of females report Fantastic also increases. The results do not report that. It might be a missing result, suggesting a heterogenous effects among students.
Author Response

(The authors gave the same response as above.)

Round 2
Reviewer 2 Report
Thank you for addressing my comments!